# Variability in the Content of Phenolic Compounds in Plum Fruit

**DOI:** 10.3390/plants9111611

**Published:** 2020-11-20

**Authors:** Mindaugas Liaudanskas, Rugilė Okulevičiūtė, Juozas Lanauskas, Darius Kviklys, Kristina Zymonė, Tamara Rendyuk, Vaidotas Žvikas, Nobertas Uselis, Valdimaras Janulis

**Affiliations:** 1Department of Pharmacognosy, Faculty of Pharmacy, Lithuanian University of Health Sciences, LT-50166 Kaunas, Lithuania; farmakog@lsmuni.lt (R.O.); valdimaras.janulis@lsmuni.lt (V.J.); 2Institute of Pharmaceutical Technologies, Faculty of Pharmacy, Lithuanian University of Health Sciences, LT-50166 Kaunas, Lithuania; kristina.zymone@lsmuni.lt (K.Z.); vaidotas.zvikas@lsmuni.lt (V.Ž.); 3Institute of Horticulture, Lithuanian Research Centre for Agriculture and Forestry, Babtai, LT-54333 Kaunas Distr., Lithuania; juozas.lanauskas@lammc.lt (J.L.); darius.kviklys@lammc.lt (D.K.); nobertas.uselis@lammc.lt (N.U.); 4Department of Horticulture, Norwegian Institute of Bioeconomy Research—NIBIO Ullensvang, Ullensvangvegen 1005, NO-5781 Lofthus, Norway; 5Departament of Natural Sciences, Institute of Pharmacy, Sechenov University, Moscow 119991, Russia; rendyuk_t_d@staff.sechenov.ru

**Keywords:** *P. cerasifera* Ehrh., *P. domestica* L., rootstock, flavonols, flavan-3-ols, chlorogenic acid, quinic acid, rutin

## Abstract

The aim of this study was to determine the composition and content of phenolic compounds in extracts of plum fruit. Fruit of 17 plum cultivars were analyzed. Fruit samples were collected in 2019 from fruit trees with “Myrobalan” (*P. cerasifera* Ehrh.) and “Wangenheim Prune” (*P. domestica* L.) rootstocks. The following glycosides of the flavonol group were identified: avicularin, isorhamnetin-3-O-rutinoside, isoquercitrin, hyperoside, rutin, and an aglycone quercetin. Compounds of the flavan-3-ol group were identified, such as (+)-catechin, procyanidin C1, and procyanidin A2, along with chlorogenic acid attributed to phenolic acids and a non-phenolic cyclitol–quinic acid. Of all the analytes identified in plum fruit samples, quinic acid predominated, while chlorogenic acid predominated among all the identified phenolic compounds, and rutin predominated in the flavonol group. Hierarchical cluster analysis (HCA) and principal component analysis (PCA) revealed that fruit samples of “Kubanskaya Kometa”, “Zarechnaya Raniaya”, “Duke of Edinburgh”, “Jubileum”, and “Favorita del Sultano” cultivars had different quantitative content of phenolic compounds from that observed in other samples. The highest total amount of phenolic compounds was found in the European plum samples of the “Zarechnaya Rannyaya” cultivar, while the amount of quinic acid was the highest in plum fruit samples of the “Jubileum” cultivar.

## 1. Introduction

Plums are among the most popular fruit grown in Lithuania and used in the food chain. Plums are used in the manufacturing of food products and are potentially valuable for the development and production of dietary supplements and functional food enriched with biologically active compounds. World plum production has been increasing slightly over the recent years. In 2018, it reached about 12.6 million tons. According to the production scale, plums were in the 7th position among the most cultivated fruit [1]. There are 19–40 species of plum, depending on taxonomies, which have originated in Europe, Asia, and America. From this great diversity, only two species—the hexaploid European plum (*Prunus domestica*) and the diploid Japanese plum (*P. salicina* and hybrids)—are of worldwide commercial significance. The origin of European plum is uncertain but may have involved *P. cerasifera* and possibly *P. spinosa* as ancestors. Japanese plums originated in China but were introduced to the West from Japan only 150 years ago [2]. Various biologically active compounds important for the vital functions of the human body have been detected in plum fruit, including flavonoids (anthocyanins, flavonols, and flavan-3-ols), phenolic acids (chlorogenic, neochlorogenic, protocatechuic, caffeic, trans-p-coumaric, and ferulic acids), and vitamins (ascorbic acid, tocopherols, phylloquinone, and carotenoids) [3,4,5]. Biologically active compounds in plum fruit have broad-spectrum biological effects, including the enhancement of cognitive functions, the reduction of the risk of cardiovascular diseases, as well as laxative and antimicrobial effects [4,6]. Qualitative and quantitative content of biologically active compounds in plum fruit may vary greatly depending on the cultivars, which highlights the importance of the evaluation of the qualitative and quantitative variability in the phytochemical composition of plum fruit of different cultivars.

The biological properties and economic importance of plum fruit trees have been generally evaluated in Lithuania. These studies highlighted the influence of cultivars and rootstocks on these parameters [7,8]. We failed to find any detailed studies on the phytochemical composition of plum fruit in scientific literature, as only fragmented studies on fruit composition are presented. This lack of the studies encouraged us to increase the scientific knowledge in the field of the studies on the detailed chemical composition of plum fruit and to evaluate the qualitative and quantitative composition of plums grown in Lithuanian climatic conditions. The obtained results would provide new information about the variability in the phytochemical composition of plum fruit grown in colder climatic conditions and would allow for identifying the most promising cultivars whose fruit accumulate the highest amounts of phenolic compounds.

The aim of this study was to evaluate the variability in the composition of phenolic compounds in plum fruit grown in Lithuania. 

## 2. Results and Discussion

### 2.1. Variability in the Qualitative and Quantitative Content of Phenolic Compounds in Plum Fruit Samples

As the chemical composition of edible fruits of different cultivars can vary considerably [5,9], it is very important to compare and assess the chemical composition of the plum fruit of different cultivars and to ensure their quality. The application of the UHPLC method and the use of methodology developed by us in the analysis of phenolic compounds allowed for the identification of the qualitative and quantitative content of individual phenolic compounds and their variability in the fruit of different plum cultivars.

We conducted a detailed analysis of the chemical composition of flavonols, flavan-3-ols, and phenolcarboxylic acids in plum fruit. During the analysis, the following glycosides of the flavonol group were detected: avicularin, isorhamnetin-3-O-rutinoside, isoquercitrin, hyperoside, and rutin and aglycone quercetin (Table 1). Scientific literature indicates that compounds of the flavonol group—especially quercetin and its glycosides—have a broad-spectrum biological effect: they reduce the risk of cardiovascular diseases [10,11], metabolic disorders [12], and certain types of cancer [13,14].

One of the flavonol group glycosides—avicularin—was detected in 8 out of 21 samples of the studied plum cultivars. The lowest amount of avicularin (3.37 ± 0.17 µg/g) was found in fruit of the “Duke of Edinburgh” (“Myrobalan” rootstock) cultivar. Small amounts of avicularin were also found in plum fruit samples of “Violeta” (10.03 ± 0.50 µg/g), “Favorita del Sultano” (14.00 ± 0.70 µg/g), and “Jubileum” (“Myrobalan” rootstock, 22.84 ± 1.14 µg/g) cultivars. The highest amount of avicularin (277.16 ± 13.86 µg/g, *p* < 0.05) was found in fruit samples of the “Kubanskaya Kometa” cultivar. The cultivar factor had the greatest (η^2^ = 0.997, *p* < 0.001) influence on the amount of avicularin in plum fruit samples. The quantitative content of this compound was significantly influenced by the rootstock factor and the interaction of the cultivar and the rootstock factors (respectively, η^2^ = 0.911 and η^2^ = 0.980, *p* < 0.001). Treutter et al. qualitatively identified avicularin in plum fruit peels, yet they did not publish the results of the quantitative analysis, even though they did mention that avicularin was found not in all samples of the studied plum cultivars [15]. Our findings confirm the results published by these researchers.

During the quantitative analysis of plum fruit samples, isorhamnetin-3-O-rutinoside was detected in samples of all cultivars, except for “Kubanskaya Kometa”. Its highest amount (65.96 ± 3.30 µg/g, *p* < 0.05) was found in the plum fruit sample of the “Jubileum” cultivar (“Wangenheim Prune” rootstock). Lower amounts of this isorhamnetin glycoside were found in plum fruit samples of “Oullins Reneklode” (0.07 ± 0.003 µg/g), “Zarechnaya Ranniaya” (2.14 ± 0.11 µg/g), “Rausvė (3.16 ± 0.16 µg/g), “Valor” (“Wangenheim Prune” rootstock 4.98 ± 0.25 µg/g), and “Valor” (“Myrobalan” rootstock 6.00 ± 0.30 µg/g) cultivars. The amount of this compound in plum fruit samples was significantly influenced by the cultivar and the rootstock factors, as well as by their interaction (respectively, η^2^ = 0.994, η^2^ = 0.926, and η^2^ = 0.949, *p* < 0.001). We did not find any literature data on the quantitative content of this compound in plum fruit samples. Treutter et al. identified isorhamnetin-3-O-glucoside in plum fruit [15], yet we did not detect this compound in the studied samples of plum cultivars, even though we did detect isorhamnetin-3-O-rutinoside and evaluated its quantitative content.

Another compound of the flavonol group—isoquercitrin—was also detected in the plum fruit samples of the studied cultivars. The highest amount of this compound (123.61 ± 6.18 µg/g, *p* < 0.05) was found in the plum fruit sample of the “Zarechnaya Ranniaya” cultivar, while the lowest amount of isoquercitrin (4.80 ± 0.24 µg/g) was detected in plum fruit samples of the “Čačanska Najbolje” (“Myrobalan” rootstock) cultivar. The amount of this compound did not differ significantly (*p* > 0.05) from that detected in plum samples of “Queen Victoria” (5.70 ± 0.28 µg/g), “Favorita del Sultano” (6.47 ± 0.32 µg/g), “Oullins Reneklode” (9.37 ± 0.47 µg/g), “Violeta” (13.39 ± 0.67 µg/g), and “Jubileum” (“Myrobalan” rootstock, 17.88 ± 0.89 µg/g) cultivars.

The amount of isoquercitrin was most influenced by the interaction of the cultivar and the rootstock factors (η^2^ = 0.986, *p* < 0.001). The separate effect of the cultivar and the rootstock factors was also significant (respectively, η^2^ = 0.974 and η^2^ = 0.980, *p* < 0.001). Treutter et al. detected isoquercitrin in all the studied plum fruit peels, and its amount ranged from 10 ± 1 µg/g to 249 ± 113 µg/g [15].

The quercetin glycoside hyperoside was detected in plum fruit samples of all cultivars, except for “Kubanskaya Kometa”. The highest amount of this compound (71.15 ± 3.56 µg/g, *p* < 0.05) was detected in the plum fruit sample of the “Zarechnaya Ranniaya” cultivar, while the lowest amount of hyperoside (4.01 ± 0.20 µg/g) was found in the sample of the “Favorita del Sultano” plum cultivar. It did not differ significantly (*p* > 0.05) from the amount of this compound found in plum fruit samples of “Violeta” (5.22 ± 0.26 µg/g), “Jubileum” (“Myrobalan” rootstock, 5.25 ± 0.26 µg/g), “Queen Victoria” (8.35 ± 0.42 µg/g), or “Herman” (11.50 ± 0.57 µg/g) cultivars.

The amount of hyperoside in the studied plum fruit samples was most affected by the cultivar factor (η^2^ = 0.987, *p* < 0.001). The influence of the rootstock factor and the interaction of the cultivar and the rootstock factors was also significant (respectively, η^2^ = 0.829 and η^2^ = 0.903, *p* < 0.001). Treutter et al. in their article indicated that the amount of hyperoside in the peels of plum fruit grown in Germany ranged from 8 ± 1.1 µg/g to 289 ± 175 µg/g [15]. Data of a study conducted in Luxembourg showed that the amount of hyperoside in plum fruit ranged from 3.88 ± 1.21 µg/g to 36.99 ± 15.43 µg/g [16]. The results of those studies corroborate the findings of our study.

Quercetin glycoside rutin is one of the most common compounds found in botanical raw material. The results of our study showed that this glycoside was the predominant compound of the flavonol group in plum fruit samples of all the studied cultivars. Its quantitative content was 52.19–89.93% of the total amount of flavonols detected in plums. The highest amount of rutin was found in plum fruit samples of “Herman” (537.81 ± 26.89 µg/g), “Rausvė” (565.85 ± 28.29 µg/g), “Stanley” (478.81 ± 23.94 µg/g), and “Zarechnaya Rannyaya” (458.39 ± 22.92 µg/g) cultivars, while the lowest amount of this compound was detected in plum fruit samples of “Violeta” (121.61 ± 6.08 µg/g), “Oullins Reneklode” (151.28 ± 7.56 µg/g), and “Opal” (190.14 ± 9.51 µg/g) cultivars.

The amount of rutin in plum fruit samples was most affected by the interaction of the cultivar and the rootstock factors (η^2^ = 0.925, *p* < 0.001). The separate effect of the cultivar and the rootstock factors was significant, albeit smaller (respectively, η^2^ = 0.663 and η^2^ = 0.618, *p* < 0.001). Slovenian researchers found in their study that the flesh of plum fruit contained 2–4 µg/g of glycoside rutin, while the peel contained 1057–1867 µg/g of this compound [17]. Treutter et al. quantitatively evaluated rutin in all the studied plum cultivars and found its amount to vary from 55 ± 27 µg/g to 1194 ± 313 µg/g [15]. Polish researchers found plum fruit samples of the “Valor” cultivar to contain 50.6 ± 0.8 µg/g of rutin, which is a smaller amount than that found in our studied sample [18]. Norwegian researchers studied plum samples of 6 cultivars and detected only minor amounts of flavonols (rutin and quercetin 3-glucoside) in all the tested cultivars [19].

The quantitative content of quercetin, an aglycone of the quercetin group glycosides, was the lowest of all the detected and quantitatively evaluated flavonol group compounds in plum fruit. This might be explained by the fact that most flavonoids are naturally accumulated in plants in the form of glycosides [20]. Quercetin was detected in plum fruit samples of only three cultivars—“Čačanska Najbolje” (“Wangenheim Prune” rootstock, 0.01 ± 0.0003 µg/g), “Zarechnaya Ranniaya” (0.19 ± 0.01 µg/g), and “Kubanskaya Kometa” (3.81 ± 0.19 µg/g). The amount of quercetin was greatly influenced by the cultivar factor and the interaction of the cultivar and the rootstock factors (respectively, η^2^ = 0.955 and η^2^ = 0.939, *p* < 0.001). The influence of the rootstock factor was not significant (*p* = 0.158). Data on the amount of this aglycone in plum fruit is lacking since, as our study showed, its amounts are small. It could be stated that this aglycone is glycosylated, and most commonly, its glycosides are detected in plum fruit samples.

The percentage part of the total amount of the identified and quantitatively evaluated compounds of the flavonol group with respect to the total amount of all the phenolic compounds found in plum fruit samples ranged from 9.6% (cultivar “Duke of Edinburgh”, vegetative rootstock) to 25.8% (cultivar “Stanley”). The lowest total amount of the identified and quantitatively evaluated compounds of the flavonol group was found in plum fruit samples of “Oullins Reneklode” (181.07 ± 9.05 µg/g), “Violeta” (160.41 ± 7.52 µg/g), “Queen Victoria” (233.62 ± 11.68 µg/g), “Opal” (251.65 ± 12.58 µg/g), “Favorita del Sultano” (262.03 ± 13.10 µg/g), and “Jubileum” (“Myrobalan” rootstock, 279.37 ± 13.97 µg/g) cultivars. The highest total amount of the compounds of this group (776.04 ± 33.89 µg/g) was found in the plum fruit sample of the “Kubanskaya Kometa” cultivar. It did not differ significantly from the amount of this compound found in plum fruit samples of “Zarechnaya Ranniaya” (712.58 ± 29.20 µg/g) or “Herman” (714.99 ± 31.04 µg/g) cultivars.

The coefficient of variation was 41.54%, which explains a significant variability in the total amount of flavonols in samples of different plum cultivars. The calculation of individual coefficients of variation for flavonols in plum samples of different cultivars yielded the following results: the coefficient of variation for isoquercitrin was 88.69%, for isorhamnetin-3-O-rutinoside—84.92%, for hyperoside—65.51%, and for rutin—38.26%.

The analysis of the compounds of the flavan-3-ol group by applying the UHPLC-ESI-MS/MS technique allowed for the identification of oligomeric procyanidins (procyanidin A2 and procyanidin C1) and a monomeric compound (+)-catechin (Table 2). The amount of procyanidin C1 ranged from 65.47 μg/g to 1484.15 ± 74.21 μg/g. The lowest amount of procyanidin C1 (65.47 ± 3.27 μg/g) was found in plum fruit samples of the “Queen Victoria” cultivar. It did not differ significantly from the amount of this compound found in plum fruit samples of “Violeta” (217.64 ± 10.88 μg/g), “Čačanska Rana” (221.59 ± 11.08 μg/g), and “Valor” (“Wangenheim Prune” rootstock, 246.43 ± 12.32 μg/g) cultivars. The highest amount of procyanidin C1 (1484.15 ± 74.21 μg/g, *p* < 0.05) was found in the plum fruit sample of the “Kubanskaya Kometa” cultivar.

The calculated coefficient of variation (62.48%) explains a significant variability in the amount of procyanidin C1 in plum fruit samples of different cultivars. The amount of procyanidin C1 was most influenced by the cultivar factor (η^2^ = 0.989, *p* < 0.001). The influence of the interaction of the cultivar and the rootstock factors on the amount of this compound in the studied plum fruit was smaller (η^2^ = 0.692, *p* < 0.001). No significant effect of the rootstock factor on the amount of procyanidin C1 was found (*p* = 0.553).

We did not find procyanidin A2, which belongs to the group of flavan-3-ols, in plum fruit samples of “Favorita del Sultano”, “Jubileum” (“Myrobalan” rootstock), “Queen Victoria”, or “Violeta” cultivars. The lowest amount of this compound was found in samples of “Stanley”, “Opal”, “Herman”, “Duke of Edinburgh” (Myrobalan rootstock), “Čačanska Rana”, “Kijevas Vela”, “Duke of Edinburgh” (“Wangenheim Prune” rootstock), “Čačanska najbolje” (“Myrobalan” rootstock), “Jubileum” (“Wangenheim Prune” rootstock), “Rausvė”, and ‘”Oullins Reneklode” cultivars, and that amount ranged from 0.03 ± 0.002 µg/g to 8.45 ± 0.42 µg/g. The highest amount of procyanidin A2 (174.44 ± 8.72 µg/g, *p* < 0.05) was found in plum fruit samples of the “Kubanskaya Kometa” cultivar. The amount of procyanidin A2 was most influenced by the cultivar factor and the interaction of the cultivar and the rootstock factors (respectively, η^2^ = 0.996 and η^2^ = 0.982, *p* < 0.001). The influence of the rootstock factor on the amount of procyanidin A2 was smaller (η^2^ = 0.493, *p* < 0.01).

The compound of the flavan-3-ol group—(+)-catechin—was detected in plum fruit samples of all the studied cultivars. Its highest amount (999.92 ± 50.00 µg/g, *p* < 0.05) was found in the plum fruit sample of the “Duke of Edinburgh” (Myrobalan rootstock) cultivar. The lowest amount (136.17 ± 6.81 µg/g) of this compound was found in plum fruit samples of the “Jubileum” (“Wangenheim Prune” rootstock) cultivar. It did not differ significantly (*p* > 0.05) from the amount of (+)-catechin found in plum fruit samples of “Rausvė” (169.39 ± 8.47 µg/g), “Jubileum” (“Myrobalan” rootstock, 222.78 ± 11.14 µg/g), and “Čačanska Rana” (271.01 ± 13.55 µg/g) cultivars. The calculated coefficient of variation (44.06%) indicates a rather high variability in the amount of (+)-catechin between plum fruit samples of different cultivars. 

The variation in the amount of the representative of the phenol group—(+)-catechin—was influenced by the cultivar factor and the interaction of the cultivar and the rootstock factors (respectively, η^2^ = 0.955 and η^2^ = 0.939, *p* < 0.001). The influence of the rootstock factor on the amount of (+)-catechin in plum fruit samples was not significant (*p* = 0.158). German researchers indicated in their publication that the amount of (+)-catechin in the peels of fresh plum fruit grown and harvested in Germany ranged from 11 ± 7 µg/g to 290 ± 101 μg/g [15]. Researchers from Luxembourg indicated in their publication that the amount of (+)-catechin in fresh plum samples ranged from 1.11 ± 0.11 μg/g to 4.66 ± 0.13 μg/g [16].

The total percentage part of the identified and quantitatively evaluated amount of flavan-3-ol group compounds ranged from 19.9% (cultivar “Herman”) to 25.8% (cultivar “Kubanskaya Kometa”) with regard to the total amount of all phenolic compounds in plum fruit samples. The lowest total amount (496.22 ± 24.81, *p* < 0.05 µg/g) of the compounds of this group was found in the plum fruit sample of the “Čačanska Rana” cultivar. It did not differ significantly from the total amount of identified flavan-3-ols found in plum fruit samples of “Valor” (“Wangenheim Prune” rootstock, 583.51 ± 29.18 µg/g), “Violeta” (613.09 ± 30.65 µg/g), “Jubileum” (“Wangenheim Prune” rootstock, 585.65 ± 29.28 µg/g), “Jubileum” (“Myrobalan” rootstock, 650.94 ± 32.55 µg/g), “Queen Victoria” (703.00 ± 35.15 µg/g), and “Herman” 770.12 ± 38.51 µg/g) cultivars. The highest amount (2210.04 ± 110.50 µg/g, *p* < 0.05)—in plum fruit samples of the “Kubanskaya Kometa” cultivar. The coefficient of variation (48.01%) showed a rather large variability in the total amount of the identified compounds of the flavan-3-ol group between plum fruit samples of different cultivars.

Spanish researchers conducted a qualitative and quantitative evaluation of the composition of plum fruit and identified more compounds of the flavan-3-ol group. They quantitatively evaluated procyanidins B1, B2, and B4 as well as A-type procyanidin dimers. The results of the study showed that that the total amount of proanthocyanidins in plum fruit peel samples ranged from 662.2 ± 133.1 μg/g to 1650.6 ± 19.6 μg/g, and in plum flesh samples—from 179.0 ± 39.7 μg/g to 566.1 ± 51.0 μg/g [21]. Comparative studies have been performed to determine variability in the amount of procyanidins in samples of other fruit. The results showed that in peach peel samples, the total amount of the compounds of the proanthocyanidin group ranged from 310.7 ± 30.0 μg/g to 1098.6 ± 120.1 μg/g, while in peach flesh samples, it ranged from 87.4 ± 15.6 µg/g to 658.5 ± 37.0 μg/g. In nectarine fruit peel samples, the total amount of the compounds of the proanthocyanidin group ranged from 93.5 ± 20.5 μg/g to 744.5 ± 143 μg/g, while in nectarine flesh samples, it ranged from 13.6 ± 8.2 µg/g to 551.0 ± 30.2 μg/g [21].

Generalizing the obtained results, in can be stated that the profiles of individual flavan-3-ols in plum fruit were quite different. Procyanidin A2 was detected only in some plum fruit samples. The total amount of the compound of the flavan-3-ol group was the highest in plum fruit samples of the “Kubanskaya Kometa” cultivar. 

Of all the identified phenolic compounds, chlorogenic acid predominated, and its amounts were significantly higher than those of other quantitatively evaluated phenolic compounds (Table 2). This acid has a wide range of potential health benefits, including its strong antioxidant [22], anti-diabetic [23,24], anti-carcinogenic [25,26], anti-inflammatory [27,28], and anti-obesity effects [29]. The lowest amount of chlorogenic acid (219.89 ± 10.99 µg/g) was detected in the plum fruit sample of the “Kubanskaya Kometa” cultivar. It did not differ significantly from the amount of this compound found in plum fruit samples of “Oullins Reneklode” (449.50 ± 22.47 µg/g), “Queen Victoria” (497.07 ± 24.85 µg/g), and “Violeta” (565.83 ± 28.29 µg/g) cultivars. The highest amount of chlorogenic acid (3126.77 ± 156.34 µg/g, *p* < 0.05) was found in the plum fruit sample of the “Zarechnaya Ranniaya” cultivar (Table 2). 

The variability in the amount of chlorogenic acid was most influenced by the cultivar factor (η^2^ = 0.978, *p* < 0.001). The evaluation showed that the interaction of the rootstock and the cultivar factors in the studied plum fruit samples affected the variability in the amount of chlorogenic acid (respectively, η^2^ = 0.755 and η^2^ = 0.885, *p* < 0.001). The percentage part of chlorogenic acid in the studied plum fruit samples ranged from 6.0% (cultivar “Kubanskaya Kometa”) to 62.3% (cultivar “Zarechnaya Ranniaya”) of the total amount of all phenolic compounds. The coefficient of variation (59.61%) showed a relatively wide variation in the amount of chlorogenic acid between plum fruit samples of different cultivars. Our results confirmed the data published by Polish researchers [18].

In this study, quinic acid, which does not belong to the group of phenolic compounds, was identified and quantitatively evaluated in plum fruit of cultivars grown in Lithuania. This acid was detected in plum fruit samples of all the studied cultivars (Table 2). Its lowest amount (3164.88 ± 158.24 µg/g) was detected in the plum fruit sample of the “Dabrowicka Prune” cultivar. It did not differ significantly from the amount of this compound found in plum fruit samples of “Čačanska Najbolje” (“Myrobalan” rootstock, 4191.15 ± 209.56 µg/g), “Čačanska Najbolje” (“Wangenheim Prune” rootstock, 4374.88 ± 218.74 µg/g), “Opal” (4590.30 ± 229.52 µg/g), and “Kubanskaya Kometa” (4828.06 ± 241.40 µg/g) cultivars. The highest amount of quinic acid was found in plum fruit samples of the “Jubileum” cultivar on the “Wangenheim Prune” rootstock and the “Myrobalan” rootstock (respectively, 8596.76 ± 429.84 and 8884.28 ± 444.21 µg/g, *p* < 0.05).

The coefficient of variation (24.74%) showed a small variability in the amount of quinic acid between plum fruit samples of different cultivars. The cultivar factor affected the variability in the amount of quinic acid between plum fruit samples (η^2^ = 0.969, *p* < 0.001). No significant effect of the rootstock factor or the interaction of the cultivar and the rootstock factors was found (*p* > 0.632). We did not find any data on the evaluation of the qualitative and quantitative content of quinic acid in plum fruit samples mentioned in scientific literature. However, scientific literature contains data about studies on chlorogenic acid and neochlorogenic acid in plum fruit samples [18,19,30].

When generalizing the obtained results, it can be stated that rutin was the predominant compound of the flavonol group in the fruit of the studied plum cultivars. In fruit samples, it comprised 52.19–89.94% of all the detected compounds of the flavonol group. The percentage amount of other flavonols in fruit samples varied depending on the cultivar. Concerning the quantitative content of flavonols compared to the amount of rutin, lower amounts of avicularin were found in fruit samples of the “Kubanskaya Kometa” cultivar, a lower amount of isorhamnetin-3-O-rutinoside was found in the fruit samples of the “Dabrowicka Prune” cultivar, a lower amount of isoquercitrin was found in the “Čačanska Najbolje” cultivar (“Wangenheim Prune” rootstock), and a lower amount of hyperoside was found in the “Čačanska Najbolje” cultivar (“Myrobalan” rootstock). Quercetin comprised the smallest percentage part of flavonols in plum fruit samples of all the studied cultivars. The results of the evaluation of flavonols confirmed those obtained by other researchers, yet there is a lack of research data on the variability of the amounts of certain compounds. The following compounds of the flavan-3-ol group were detected in plum fruit samples: (+)-catechin, procyanidin C1, and procyanidin A1. The quantitative content of the compounds of this group depended on the cultivar factor. In our studied plum cultivar samples, either (+)-catechin or procyanidin C1 predominated. Chlorogenic acid was detected in plum samples of all the studied cultivars, and its quantitative content was the highest of all the identified and quantitatively evaluated phenolic compounds in plum samples of our studied cultivars. Chlorogenic acid also predominated in fruit of plants belonging to the *Rosaceae* Juss. family [30].

### 2.2. Comparison of the Plum Fruit Samples Using Cluster Analysis 

Hierarchical analysis was applied based on the mean quantities of phenolic compounds as clustering variables. The clustering of the plum fruit resulted in grouping of the samples into five main clusters (Figure 1). 

Quinic and chlorogenic acids can be regarded as the principal markers in plum fruit, followed by rutin, catechin, and procyanidin C1 from the flavonoid complex (Figure 2). The first cluster grouped samples of “Valor” (both rootstocks), “Čačanska Rana”, “Stanley”, “Violeta”, “Queen Victoria”, “Oullins Reneklode”, “Opal”, “Čačanska Najbolje” (both rootstocks), “Kijevas Vela”, and “Dabrowicka Prune”. The corresponding samples differed from the others by the lowest contents of quercetin, avicularin, isoquercitrin, rutin, isorhamnetin 3-O-rutinoside, procyanidin A2, and quinic acid. The samples of “Duke of Edinburgh” (both rootstocks), “Jubileum” (both rootstocks), and “Favorita del Sultano” formed the second cluster. The cluster was characterized by the lowest contents of rutin, avicularin, isoquercitrin, procyanidin C1, and procyanidin A2, a low content of hyperoside, and the highest contents of isorhamnetin 3-O-rutinoside and quinic acid. Fruit of “Herman” and “Rausve” cultivars were grouped into the third cluster. These samples differed from the others by the lowest contents of catechin, isoquercitrin, isorhamnetin 3-O-rutinoside, procyanidin C1, procyanidin A2, and quinic acid, and the highest contents of rutin. Fruits of “Zarechnaya Ranniaya” and ‘Kubanskaya Kometa” cultivars formed the fourth and the fifth clusters, respectively. Fruit samples of the “Zarechnaya Ranniaya” cultivar differed from the others by the highest contents of chlorogenic acid, hyperoside, isoquercitrin, and rutin, and the lowest contents of isorhamnetin 3-O-rutinoside, procyanidin C1, procyanidin A2, and quinic acid. The samples of “Kubanskaya Kometa” fruits were characterized by the highest contents of quercetin, avicularin, procyanidin C1, and procyanidin A2, and the lowest contents of chlorogenic and quinic acids. 

A principal component analysis (PCA) was performed to detect similarities and differences between the analyzed samples according to statistically independent variables, which indicate the content of biologically active compounds. Figure 2 summarizes the PCA results based on the correlation matrix with PC1, PC2, and PC3, which explain 76.84% of the total variance in the plum fruit data sets. The score plot models for fruit samples have shown relatively good separation between the plum cultivars (Figure 2).

In the PCA model, PC1 described 37.53% of the total variance of data and highly correlated with positive loadings of quercetin (0.970), avicularin (0.944), procyanidin A2 (0.932), catechin (0.707), and procyanidin C1 (0.696). PC2 accounted for 22.96% of the total variance and was characterized by the positive loadings of chlorogenic acid (0.847), hyperoside (0.781), isoquercitrin (0.739), and rutin (0.685). PC3 described 16.35% of the observed total variability and was associated with positive loadings of quinic acid (0.791) and isorhamnetin 3-O-rutinoside (0.790). The PCA score plots of fruit samples showed their arrangement into four distinct groups (Figure 2). The fruit of “Kubanskaya Kometa” cultivar were at a distance from all the others. The clustering of this sample along the positive PC1 can be explained by the highest values of quercetin, avicularin catechin, procyanidin A2, and procyanidin C1. Significant contents of chlorogenic acid, hyperoside, isoquercitrin, and rutin scoring high in PC2 were found in fruit of the “Zarechnaya Raniaya” cultivar. On the other hand, fruits of “Duke of Edinburgh” (both rootstocks), “Jubileum” (both rootstocks), and “Favorita del Sultano” cultivars demonstrated a close position in the PC1 vs. PC3 space scoring high in PC3, which suggested high contents of quinic acid and isorhamnetin 3-O-rutinoside in the corresponding samples. Fruit of “Valor” (both rootstocks), “Čačanska Rana”, “Stanley”, “Violeta”, “Queen Victoria”, “Oullins Reneklode”, “Opal”, “Čačanska Najbolje” (both rootstocks), “Kijevas Vela”, and “Dabrowicka Prune” cultivars demonstrated a close position in the PC1 vs. PC3 space indicating their similarity in the content of phenolic compounds. Fruit samples of these cultivars clustered closely near the zero point of PC1 and PC3, indicating that the contents of flavan-3-ols, quercetin, avicularin, isorhamnetin 3-O-rutinoside, and quinic acid were found in the range of the lowest to the mean values.

As a result, PCA revealed that “Kubanskaya Kometa”, “Zarechnaya Raniaya”, “Duke of Edinburgh” (both rootstocks), “Jubileum” (both rootstocks), and “Favorita del Sultano” fruit samples had different quantitative content of phenolic compounds from that in other plum cultivars. Meanwhile, “Valor” (both rootstocks), “Čačanska Rana”, “Stanley”, “Violeta”, “Queen Victoria”, “Oullins Reneklode”, “Opal”, “Čačanska Najbolje” (both rootstocks), “Kijevas Vela”, and “Dabrowicka Prune” were similar in their phytochemical profiles. 

## 3. Materials and Methods

### 3.1. Plant Material and Climate Conditions

Samples of the following plum cultivars were included in the study: “Čačanska Najbolje”, “Čačanska Rana”, “Dąbrowicka Prune”, “Duke of Edinburgh”, “Favorita del Sultano”, “Herman”, “Jubileum”, “Kijevas Vēlā”, “Kubanskaya Kometa”, “Opal”, “Rausvė”, “Stanley”, “Oullins Reneklode”, “Valor”, “Queen Victoria”, “Violeta”, and “Zarechnaya Rannyaya”. Of these, “Čačanska Najbolje”, “Duke of Edinburgh”, “Jubileum”, and “Valor” cultivars were grafted on two rootstocks—“Myrobalan” (*P. cerasifera* Ehrh.) and “Wangenheim Prune” (*P. domestica* L.), and the rest were grafted on “Myrobalan” rootstocks. The plum orchard was planted at the Institute of Horticulture, Lithuanian Research Centre for Agriculture and Forestry (55°60′ N, 23°48′ E), in 2012. Fruit trees were planted in a randomized block design, with four replicates and three trees per plot. 

Climate conditions in 2019. Lower than Standard Climate Rate (SCR) temperatures and low precipitation rate in the first half of May resulted in spring frosts during two nights, but it did not have negative influence on fruit set and fruit growth. Despite of drought conditions in June due to very hot weather (3.2–6.6 °C higher than SCR) and lower precipitation rate, plum fruit trees did not exhibited stress and fruit growth rate was close to optimal. Temperatures and precipitation rate in July, August and in the first half of September were close to multi-year average and were favourable for optimal plum fruit and tree development.

### 3.2. Chemicals

All the solvents, reagents, and standards used were of analytical grade. The following substances were used in the study: ethanol 96% (**v/v**) (AB “Vilniaus degtinė”, Vilnius, Lithuania), 4-(dimethylamino)cinnamaldehyde (99% purity), acetic acid (99.8% purity) (“Scharlau”, Sentmenat, Spain), procyanidin A2 (99% purity), procyanidin C1 (90% purity), quercetin (95% purity), quinic acid (98% purity), hyperoside (97% purity), isorhamnetin-3-O-rutinoside (95% purity), avicularin (90% purity), formic acid (97.5–98.5% purity), acetonitrile (99.9% purity), (+)-catechin (98% purity), rutin (95% purity), isoquercitrin (90% purity), chlorogenic acid (95% purity), and hydrochloric acid (37% purity) (Sigma-Aldrich, Steinheim, Germany). During the study, we used purified de-ionized water prepared with the “Milli–Q^®^” (“Millipore”, Bedford, MA, USA) water purification system.

### 3.3. Preparation of the Plum Fruit Sample Extracts 

#### 3.3.1. Preparation of the Raw Material

For the analysis, we selected 20 ripe plum fruit from each cultivar. The plum fruit were pitted, and the remaining unpeeled parts of the fruit were cut into up to 1-cm slices and were frozen (at −35 °C) in a freezer with air circulation. Plum fruit were dried in a lyophilizer “Zirbus” (“Zirbus technology GmbH”, Bad Grund, Germany) with a pressure of 0.01 mbar at the condenser temperature of −85 °C. The lyophilized fruit were then ground using an electrical mill “Retsch GM 200” (Retsch GmbH, Haan, Germany) and were stored in tightly closed vessels in a dark and dry place. The results of the analysis were calculated for the absolute dry weight of raw material.

#### 3.3.2. Preparation of the Studied Plum Fruit Extracts

The ground raw material and reagents were weighed by using electronic analytical scales “Sartorius CP64–0CE” (“Sartorius AG”, Göttingen, Germany). For the analysis, 2.5 g of lyophilized plum fruit powder was poured into a dark glass vial, subsequently adding 50 mL of 83.64% (*v/v*) ethanol with 0.1% hydrochloric acid. The extraction of plum fruit samples was carried out in a “Bandelin Sonorex Digital 10 P” (“Bandelin Electronic GmbH & Co. KG”, Darmstadt, Germany) ultrasonic bath. The extraction was carried out for 40 min at 80 Hz frequency and 678 W power. After the extraction, the samples were centrifuged using a “Heraeus Biofuge Stratos” (“Heraeus Holding GmbH”, Haan, Germany) centrifuge for 5 min at 9000 rpm at room temperature. The supernatant was then poured off the precipitations, filtered through cotton wool, and was poured into dark glass vials, which were stored in a fridge at 4 °C until the analysis. Prior to chromatography, the extracts were additionally filtered through membrane filters with 0.22 μm pore size.

### 3.4. Evaluation of Phenolic Compounds in Plum Samples Using the UHPLC-ESI-MS/MS Technique

The variability in the qualitative and quantitative content of phenolic compounds in plum fruit samples was evaluated by applying ultra-high performance liquid chromatography (UHPLC) mass spectrometry, using a technique described and validated in an article by Gonzalez-Burgos et al. [30]. The analysis of the qualitative and quantitative content of phenolic compounds in the samples of the fruit extracts of the studied plum cultivars was carried out using a liquid chromatography system “Waters ACQUITY UPLC^®^ H–Class” (“Waters”, Milford, MA, USA) with a tandem quadrupole mass detector “Xevo TQD” (Waters, Milford, MA, USA). Sorting out of the compounds was performed using a “YMC Triart C18” (100 Å, 100 × 2.0 mm; particle size 1.9 μm) column (“YMC”, Kyoto, Japan) with a pre-column. The mass spectrometry parameters for the analysis of phenolic compounds and quinic acid are presented in Table 3.

### 3.5. Data Analysis

Statistical analysis of the data was carried out using “Microsoft Office Excel 2013” (“Microsoft”, Redmond, WA, USA) and “SPSS 20.0” (“SPSS Inc.”, Chicago, IL, USA) computer software. During the study, we calculated arithmetic means and standard deviations of three repeated measurements. Variability in the quantitative content was evaluated by calculating the coefficient of variation. A univariate dispersion analysis model (One-Way ANOVA) using Tukey’s multiple comparison test was applied for determining statistically significant differences between different plum cultivars. In order to evaluate the influence of the cultivars and the rootstock and a possible interaction of these factors, two-way ANOVA was applied. Differences were regarded as statistically significant when *p* < 0.05. According to the quantitative content of the identified compounds, the tested samples were compared by the method of cluster analysis using squared Euclidean distances. Principal component analysis (PCA) was performed taking into account factors with eigenvalues higher than 1.

## 4. Conclusions

In conclusion, the results of this study will provide new knowledge about the composition and content of phenolic compounds in European plum fruit, which will give a wide range of possibilities to employ these plants as a source of phenolic compounds. “Kubanskaya Kometa”, “Zarechnaya Raniaya”, “Duke of Edinburgh”, “Jubileum”, and “Favorita del Sultano” cultivar were characterized by an exclusive phytochemical composition. The highest total amount of the identified phenolic compounds was found in European plum samples of the “Zarechnaya Ranniaya” cultivar, while the highest amount of quinic acid was found in plum fruit samples of the “Jubileum” cultivar. Of all the identified phenolic compounds, chlorogenic acid predominated, and its amounts were significantly higher than those of other quantitatively evaluated phenolic compounds. The percentage part of chlorogenic acid in the studied plum fruit samples comprised 6.0% to 62.3% of the total amount of all phenolic compounds. Of all the identified analytes, quinic acid, which does not belong to the group of phenolic compounds, stood out, as its amount in plum fruit samples of some cultivars was even higher than the total amount of the identified phenolic compounds. 

The results of the conducted evaluations of the variability in the composition of phenolic compounds and quinic acid in plum fruit samples of different cultivars are relevant and valuable both in the theoretical and the practical aspects. They allow for the selection of the most promising plum cultivars, helping to provide the customers with high-quality plum fruit with a known composition, containing the highest accumulated amounts of phenolic compounds, which are natural antioxidants. Plum fruit were found to contain phenolic compounds of different groups with a wide-range biological effect. For this reason, plum fruit have a potential value for the needs of practical medicine, including the isolation of individual compounds with a specific effect and the development and production of preventive phytopreparations, dietary supplements, and functional food.

## Figures and Tables

**Figure 1 plants-09-01611-f001:**
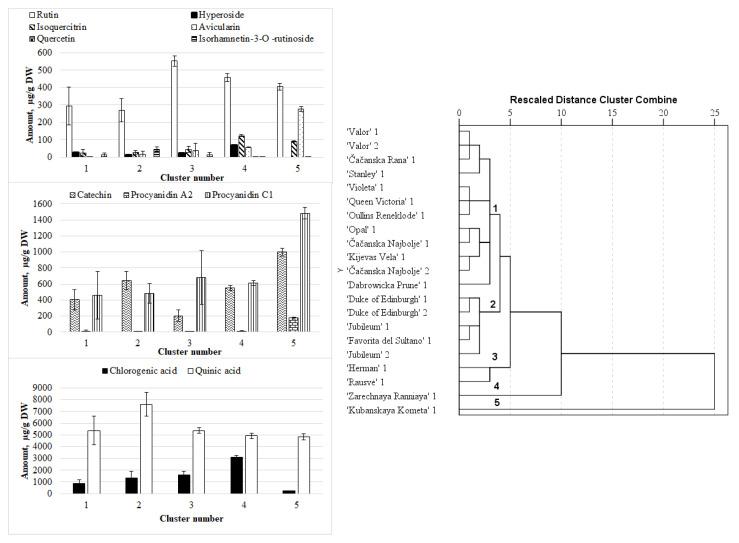
The dendrogram of hierarchical cluster analysis (HCA) of plum fruit samples based on the phytochemical composition and mean values of contents of the identified compounds (μg/g DW) of plum cultivar fruit groups extracted using HCA. 1—“Myrobalan” (*P. cerasifera* Ehrh.) rootstock; 2—“Wangenheim Prune” (*P. domestica* L.) rootstock.

**Figure 2 plants-09-01611-f002:**
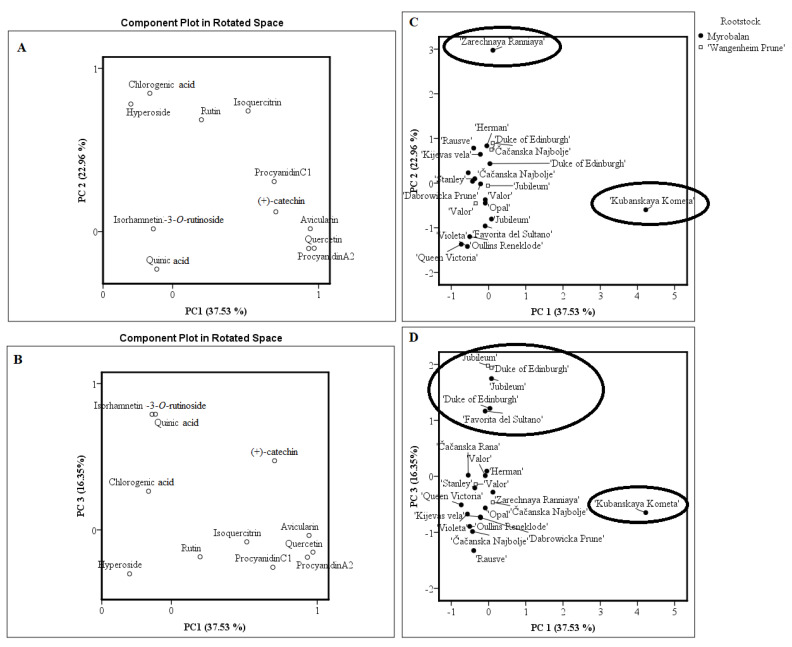
PCA loading (**A**,**B**) and score (**C**,**D**) plots of plum fruit samples of different cultivars.

**Table 1 plants-09-01611-t001:** Variability in the quantitative content (µg/g DW) of identified flavonols in plum fruit samples. The different letters indicate statistically significant (*p* < 0.05) differences between the quantitative values of flavonols in plum fruit samples of different cultivars. 1—“Myrobalan” (*P. cerasifera* Ehrh.) rootstock; 2—“Wangenheim Prune” (*P. domestica* L.) rootstock. ND—not detected.

Amount, µg/g DW	Avicularin	Hyperoside	Isoquercitrin	Isorhamnetin-3-O-Rutinoside	Quercetin	Rutin	Total Flavonols
“Čačanska Najbolje” 1	ND	40.53 ± 2.03 ^c,d^	4.80 ± 0.24 ^i^	14.74 ± 0.74 ^g^	ND	265.53 ± 13.28 ^g,h,i,j^	325.61 ± 16.28 ^g,h,i,j^
“Čačanska Najbolje” 2	ND	38.62 ± 1.93 ^c,d^	63.53 ± 3.18 ^c^	17.35 ± 0.87 ^g^	0.01 ± 0.0003 ^b^	358.98 ± 17.95 ^c,d,e,f,g^	478.48 ± 23.92 ^d,e,f^
“Čačanska Rana” 1	ND	42.80 ± 2.14 ^c^	30.79 ± 1.54 ^d,e,f^	15.37 ± 0.77 ^g^	ND	359.04 ± 18.07 ^c,d,e,f,g^	448.00 ± 22.40 ^d,e,f,g^
“Dąbrowicka Prune” 1	ND	22.56 ± 1.13 ^f^	16.57 ± 0.83 ^g,h,i^	43.44 ± 2.17 ^b,c^	ND	432.60 ± 21.63 ^b,c,d,e^	515.17 ± 25.76 ^c,d^
“Duke of Edinburgh” 1	3.37 ± 0.17 ^e^	21.59 ± 1.08 ^f^	39.60 ± 1.98 ^d^	39.87 ± 1.99 ^c,d^	ND	335.38 ± 16.77 ^d,e,f,g^	439.81 ± 21.99 ^d,e,f,g^
“Duke of Edinburgh” 2	ND	21.48 ± 1.18 ^f^	38.38 ± 1.92 ^d^	48.03 ± 2.40 ^b^	ND	280.91 ± 14.05 ^f,g,h,i^	388.81 ± 19.44 ^e,f,g,h^
“Favorita del Sultano” 1	14.00 ± 0.70 ^e^	4.01 ± 0.20 ^h^	6.47 ± 0.32 ^i^	42.15 ± 2.11 ^b,c,d^	ND	195.40 ± 9.77 ^i,j^	262.03 ± 13.10 ^i,j,k^
“Herman” 1	76.64 ± 3.83 ^b^	11.50 ± 0.57 ^g,h^	61.99 ± 3.10 ^c^	27.05 ± 1.35 ^e,f^	ND	537.81 ± 26.89 ^a,b^	714.99 ± 31.04 ^a,b^
“Jubileum” 1	22.84 ± 1.14 ^d,e^	5.25 ± 0.26 ^h^	17.88 ± 0.89 ^f,g,h,i^	34.79 ± 1.74 ^d,e^	ND	198.60 ± 9.93 ^i,j^	279.37 ± 13.97 ^h,i,j,k^
“Jubileum” 2	44.77 ± 2.24 ^c,d^	20.43 ± 1.02 ^f^	28.74 ± 1.44 ^d,e,f,g^	65.96 ± 3.30 ^a^	ND	337.84 ± 16.89 ^d,e,f,g^	497.75 ± 24.89 ^d,e^
“Kijevas Vēlā” 1	ND	53.49 ± 2.67 ^b^	71.48 ± 3.57 ^c^	12.38 ± 0.62 ^g,h^	ND	350.25 ± 17.51 ^d,e,f,g^	487.60 ± 24.38 ^d,e,f^
“Kubanskaya Kometa” 1	277.16 ± 13.86 ^a^	ND	90.03 ± 4.50 ^b^	ND	3.81 ± 0.19 ^a^	405.05 ± 20.25 ^c,d,e,f^	776.04 ± 33.89 ^a^
“Opal” 1	ND	20.30 ± 1.07 ^f^	31.29 ± 1.56 ^d,e^	9.91 ± 0.50 ^g,h,i^	ND	190.14 ± 9.51 ^i,j,k^	251.65 ± 12.58 ^i,j,k^
“Oullins Reneklode” 1	ND	20.35 ± 1.02 ^f,g^	9.37 ± 0.47 ^h,i^	0.07 ± 0.003 ^j^	ND	151.28 ± 7.56 ^j,k^	181.07 ± 9.05 ^k^
“Queen Victoria” 1	ND	8.35 ± 0.42 ^h^	5.70 ± 0.28 ^i^	9.46 ± 0.47 ^g,h,i^	ND	210.10 ± 10.51 ^h,i,j^	233.62 ± 11.68 ^j,k^
“Rausvė” 1	ND	37.57 ± 1.88 ^c,d^	29.50 ± 1.48 ^d,e,f,g^	3.16 ± 0.16 ^i,j^	ND	565.85 ± 28.29 ^a^	636.08 ± 31.80 ^b,c^
“Stanley” 1	ND	31.98 ± 1.60 ^d,e^	14.90 ± 0.75 ^h,i^	26.24 ± 1.31 ^f^	ND	478.81 ± 23.94 ^a,b,c^	551.93 ± 27.60 ^c,d^
“Valor” 1	ND	21.60 ± 1.14 ^f^	13.73 ± 0.69 ^h,i^	6.00 ± 0.30 ^h,i,j^	ND	325.83 ± 16.29 ^e,f,g,h^	367.17 ± 18.36 ^f,g,h,i^
“Valor” 2	ND	27.72 ± 1.39 ^e,f^	22.58 ± 1.13 ^e,f,g,h^	4.98 ± 0.25 ^h,i,j^	ND	272.67 ± 13.63 ^g,h,i,j^	327.94 ± 16.40 ^g,h,i,j^
“Violeta” 1	10.03 ± 0.50 ^e^	5.22 ± 0.26 ^h^	13.39 ± 0.67 ^h,i^	10.16 ± 0.51 ^g,h,i^	ND	121.61 ± 6.08 ^k^	160.41 ± 7.52 ^k^
“Zarechnaya Rannyaya” 1	57.10 ± 2.85 ^b,c^	71.15 ± 3.56 ^a^	123.61 ± 6.18 ^a^	2.14 ± 0.11 ^i,j^	0.19 ± 0.01 ^b^	458.39 ± 22.92 ^a,b,c,d^	712.58 ± 29.20 ^a,b^

**Table 2 plants-09-01611-t002:** Variability in the quantitative content (µg/g DW) of identified flavan-3-ols, chlorogenic and quinic acids in plum fruit samples; different letters indicate statistically significant (*p* < 0.05) differences between the quantitative values of these compounds in plum fruit samples of different cultivars. 1—“Myrobalan” (*P. cerasifera* Ehrh.) rootstock; 2—"Wangenheim Prune” (*P. domestica* L.) rootstock. ND—not detected.

Amount, µg/g DW	(+)-Catechin	Procyanidin A2	Procyanidin C1	Total flavan-3-ols	Chlorogenic Acid	Quinic Acid
“Čačanska Najbolje” 1	464.07 ± 23.20 ^e,f,g,h^	5.65 ± 0.28 ^d,e,f^	852.78 ± 42.64 ^b,c^	1322.51 ± 66.13 ^c,d^	1265.66 ± 63.28 ^d,e,f^	4191.15 ± 209.56 ^g,h^
“Čačanska Najbolje” 2	674.76 ± 33.74 ^b,c^	14.69 ± 0.73 ^c,d,e^	946.21 ± 47.31 ^b,c^	1635.67 ± 81.78 ^b,c^	1382.33 ± 69.12 ^d,e^	4374.88 ± 218.74 ^g,h^
“Čačanska Rana” 1	271.01 ± 13.55 ^i,j,k,l^	3.62 ± 0.18 ^e,f^	221.59 ± 11.08 ^l,m^	496.22 ± 24.81 ^j^	1132.46 ± 56.62 ^e,f,g^	6902.54 ± 345.13 ^b,c,d^
“Dąbrowicka Prune” 1	540.25 ± 27.01 ^c,d,e,f^	18.17 ± 0.91 ^c,d^	791.42 ± 39.57 ^c,d,e^	1349.83 ± 67.49 ^b,c,d^	750.00 ± 37.50 ^h,i,j,k^	3164.88 ± 158.24 ^h^
“Duke of Edinburgh” 1	999.92 ± 50.00 ^a^	3.36 ± 0.17 ^e,f^	665.58 ± 33.28 ^d,e,f^	1668.85 ± 83.44 ^b^	1573.96 ± 78.70 ^c,d^	6947.93 ± 347.40 ^b,c,d^
“Duke of Edinburgh” 2	345.80 ± 17.29 ^h,i,j^	4.65 ± 0.23 ^e,f^	556.00 ± 27.80 ^f,g,h^	906.45 ± 45.32 ^e,f,g,h^	2285.41 ± 114.27 ^b^	7002.16 ± 350.11 ^b,c,d^
“Favorita del Sultano” 1	640.10 ± 32.00 ^c,d^	ND	324.69 ± 16.23 ^j,k,l^	964.79 ± 48.24 ^e,f,g^	803.27 ± 40.16 ^g,h,i,j,k^	6608.52 ± 330.43 ^c,d,e^
“Herman” 1	394.65 ± 19.73 ^f,g,h,i^	2.95 ± 0.15 ^e,f^	372.53 ± 18.63 ^i,j,k,l^	770.12 ± 38.51 ^f,g,h,i,j^	1887.55 ± 94.38 ^c^	5370.40 ± 268.52 ^c,d,e,f,g^
“Jubileum” 1	222.78 ± 11.14 ^j,k,l^	ND	428.16 ± 21.41 ^h,i,j,k^	650.94 ± 32.55 ^g,h,i,j^	918.21 ± 45.91 ^f,g,h,i,j^	8884.28 ± 444.21 ^a^
“Jubileum” 2	136.17 ± 6.81 ^l^	5.86 ± 0.29 ^d,e,f^	443.62 ± 22.18 ^g,h,i,j^	585.65 ± 29.28 ^h,i,j^	1017.72 ± 50.89 ^e,f,g,h^	8596.76 ± 429.84 ^a,b^
“Kijevas Vēlā” 1	798.74 ± 39.94 ^b^	3.94 ± 0.20 ^e,f^	532.41 ± 26.62 ^f,g,h,i^	1335.08 ± 66.75 ^c,d^	634.62 ± 31.73 ^i,j,k^	4994.32 ± 249.72 ^e,f,g^
“Kubanskaya Kometa” 1	551.44 ± 27.57 ^c,d,e^	174.44 ± 8.72 ^a^	1484.15 ± 74.21 ^a^	2210.04 ± 110.50 ^a^	219.89 ± 10.99 ^l^	4828.06 ± 241.40 ^f,g,h^
“Opal” 1	505.08 ± 25.25 ^d,e,f^	0.36 ± 0.02 ^f^	798.86 ± 39.94 ^c,d^	1304.30 ± 65.22 ^d^	936.36 ± 46.82 ^f,g,h,i,j^	4590.30 ± 229.52 ^g,h^
“Oullins Reneklode” 1	584.39 ± 29.22 ^c,d,e^	8.45 ± 0.42 ^d,e,f^	274.90 ± 13.75 ^j,k,l^	867.74 ± 43.39 ^e,f,g,h,i^	449.50 ± 22.47 ^k,l^	5438.90 ± 271.95 ^c,d,e,f,g^
“Queen Victoria” 1	637.53 ± 31.88 ^c,d^	ND	65.47 ± 3.27 ^m^	703.00 ± 35.15 ^f,g,h,i,j^	497.07 ± 24.85 ^k,l^	6346.30 ± 317.31 ^c,d,e,f^
“Rausvė” 1	169.39 ± 8.47 ^k,l^	8.00 ± 0.40 ^d,e,f^	986.80 ± 49.34 ^b^	1164.19 ± 58.21 ^d,e^	1349.01 ± 67.45 ^d,e^	5353.67 ± 267.68 ^d,e,f,g^
“Stanley” 1	519.88 ± 25.99 ^d,e,f^	0.03 ± 0.002 ^f^	316.86 ± 15.84 ^j,k,l^	836.77 ± 41.84 ^f,g,h,i^	748.53 ± 37.42 ^h,i,j,k^	5028.95 ± 251.45 ^e,f,g^
“Valor” 1	503.22 ± 25.16 ^d,e,f,g^	49.80 ± 2.49 ^b^	277.04 ± 13.85 ^j,k,l^	830.07 ± 41.50 ^f,g,h,i^	1107.73 ± 55.39 ^e,f,g,h^	6925.10 ± 346.25 ^b,c,d^
“Valor” 2	310.22 ± 15.51 ^i,j,k^	26.86 ± 1.34 ^c^	246.43 ± 12.32 ^k,l,m^	583.51 ± 29.18 ^i,j^	972.68 ± 48.63 ^f,g,h,i^	7083.44 ± 354.17 ^b,c^
“Violeta” 1	395.45 ± 19.77 ^f,g,h,i^	ND	217.64 ± 10.88 ^l,m^	613.09 ± 30.65 ^h,i,j^	565.83 ± 28.29 ^j,k,l^	5304.72 ± 265.24 ^d,e,f,g^
“Zarechnaya Rannyaya” 1	355.63 ± 17.78 ^g,h,i,j^	13.55 ± 0.68 ^d,e^	612.80 ± 30.64 ^e,f,g^	981.98 ± 49.10 ^e,f^	3126.77 ± 156.34 ^a^	4911.80 ± 245.59 ^e,f,g^

**Table 3 plants-09-01611-t003:** Mass spectrometry parameters for the analysis of phenolic compounds and quinic acid.

Compound	Parent Ion (m/z)	Daughter Ion (m/z)	Cone Voltage, V	Collision Energy, eV
Quinic acid	191	85	40	26
(+)-Catechin	289	123	60	34
Quercetin	301	151	48	20
Chlorogenic acid	353	191	32	14
Avicularin	433	301	50	20
Hyperoside	463	300	50	26
Isoquercitrin	463	301	52	28
Procyanidin A2	575	285	50	25
Rutin	609	300	70	38
Isorhamnetin-3-O-rutinoside	623	315	70	32
Procyanidin C1	865.2	125	56	60

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
