# Peer review of "Variability in the Content of Phenolic Compounds in Plum Fruit"

_plants, 2020, doi:10.3390/plants9111611_

Round 1

Reviewer 1 Report

Article entitled “Variability in the content of phenolic compounds in plum fruit” can be considered for publication in "Plants" MDPI journal but following changes should be done

Figures 1-4 should be changed in 1 or 2 tables.
In Material and methods section author should add information concerning the climate during the growing of plums.
Line 388 subchapter 3.3 Equipment should be removed. All information about equipment used in research should be put in proper subchapters.
Line 408. Authors have stated that for freezing samples they used freezer with air circulation. It could caused the changes in polyphenolic compounds content. Please explain it.
Line 413 for extract preparation the 83.64% (v/v) ethanol with 0.1% hydrochloric acid was used. Please explain why this solution was used? Based on authors experience or literature review?
In Results and discussion climate condition (temperature, rainfall, etc.) should be discussed based on received results.

Author Response

Dear Reviewer, 

thank you for Your time devoted for our scientific work and suggestions how to improve this manuscript. Please find our explanations to Your questions and remarks in the attach file. We strongly believe that our manuscript is suitable for publication in Plants journal and this special issue.

On behalf of all the authors,

Yours sincerely,

Dr. Mindaugas Liaudanskas

Reviewer 2 Report

The results of this study will provide new knowledge about the composition and content of phenolic compounds in European plum fruit, which will give a wide range of possibilities to employ these plants as a source of phenolic compounds. The results of the conducted evaluations of the variability in the composition of phenolic compounds and quinic acid in plum fruit samples of different cultivars are relevant and valuable both in the theoretical and the practical aspects. Plum fruit were found to contain phenolic compounds of different groups with a wide range of potential biological effects for human consumption. Interesting scientific research.

Author Response

Dear Reviewer, 

thank you for Your time devoted for this scientific work. We strongly believe that our manuscript is suitable for publication in Plants journal and this special issue.

On behalf of all the authors,

Yours sincerely,

Dr. Mindaugas Liaudanskas

Reviewer 3 Report

This manuscript treats flavonoid constituent profiles of plum samples mainly. The experiment design was good, and the manuscript was written well, although the following minor points should be mentioned by the authors.

1) The wavelength used for the quantitative analysis with HPLC should be given. Maybe based on mass chromatogram?

2) Quinic acid has poor UV absorption. How did the authors quantify it?

3) Co-existence of procyanidin B2 and related compounds with procyanidin C1 has been reported by many researchers. This manuscript also mentioned those dimeric procyanidins of this plant. However, quantitative data for these dimers were not given in the manuscript. Why?

4) Purity of the standard compounds used for the quantitative analysis was not given. However, it is important for the quantitative view.

Author Response

Dear Reviewer, 

thank you for Your time devoted for our scientific work and suggestions how to improve this manuscript. Please find our explanations to Your questions and remarks in the attached file. We strongly believe that our manuscript is suitable for publication in Plants journal and this special issue.

On behalf of all the authors,

Yours sincerely,

Dr. Mindaugas Liaudanskas
